# Evaluation of the Inhibitory Potential of Synthetic Peptides Homologous to CDR3 Regions of a Monoclonal Antibody against Bothropic Venom Serine Proteases

**DOI:** 10.3390/ijms25105181

**Published:** 2024-05-09

**Authors:** Lucas Yuri Saladini, Marcos Jorge Magalhães-Junior, Cristiane Castilho Fernandes da Silva, Priscila Gonçalves Coutinho Oliveira, Roberto Tadashi Kodama, Lais Gomes, Milton Yutaka Nishiyama-Jr, Patrick Jack Spencer, Wilmar Dias da Silva, Fernanda Calheta Vieira Portaro

**Affiliations:** 1Laboratory of Structure and Function of Biomolecules, Butantan Institute, São Paulo 05503-900, Brazilcrisscastilho@gmail.com (C.C.F.d.S.); priscila.oliveira.esib@esib.butantan.gov.br (P.G.C.O.); pararoberval@gmail.com (R.T.K.); lais.gomes@icb.usp.br (L.G.); 2Laboratory of Immunochemistry, Butantan Institute, São Paulo 05503-900, Brazil; 3Laboratory of Applied Toxinology, Center of Toxins, Immune-Response and Cell Signaling (CeTICS), Instituto Butantan, São Paulo 05503-900, Brazil; milton.nishiyama@butantan.gov.br; 4Biotechnology Center, Nuclear and Energy Research Institute (IPEN/CNEN/SP), São Paulo 05503-900, Brazil; pspencer@ipen.br

**Keywords:** snakebites, antibodies, antigen, inhibitors, peptides

## Abstract

Snakebite accidents, neglected tropical diseases per the WHO, pose a significant public health threat due to their severity and frequency. Envenomation by *Bothrops* genus snakes leads to severe manifestations due to proteolytic enzymes. While the antibothropic serum produced by the Butantan Institute saves lives, its efficacy is limited as it fails to neutralize certain serine proteases. Hence, developing new-generation antivenoms, like monoclonal antibodies, is crucial. This study aimed to explore the inhibitory potential of synthetic peptides homologous to the CDR3 regions of a monoclonal antibody targeting a snake venom thrombin-like enzyme (SVTLE) from *B. atrox* venom. Five synthetic peptides were studied, all stable against hydrolysis by venoms and serine proteases. Impressively, four peptides demonstrated uncompetitive SVTLE inhibition, with Ki values ranging from 10^−6^ to 10^−7^ M. These findings underscore the potential of short peptides homologous to CDR3 regions in blocking snake venom toxins, suggesting their promise as the basis for new-generation antivenoms. Thus, this study offers potential advancements in combatting snakebites, addressing a critical public health challenge in tropical and subtropical regions.

## 1. Introduction

Snakebites, classified by the World Health Organization (WHO) as a neglected tropical disease, pose a public health concern due to their prevalence in tropical and subtropical regions [1]. Globally, approximately 5 million snakebite incidents occur each year, and about 100,000 fatalities [1,2]. Among the various snake genera, *Bothrops* stands out for its high mortality rate, being responsible for the highest number of envenoming cases in Central and South America [3,4]. Notably, the *Bothrops atrox* snake is particularly noteworthy in this regard, due to the high number of accidents caused by the species and its geographical distribution, being the main viperid present in the Amazon region [5,6]. According to proteomic data, *Bothrops atrox* venom, like other bothropic venoms, comprises a complex mixture of several components, such as metalloproteases (SVMPs, Snake Venom Metalloproteinases), serine proteases (SVSPs, Snake Venom Serine Proteases), phospholipase A^2^, C-type lectins, cysteine-rich secretory proteins (CRISPs), and L-amino acid oxidases, among others [7,8]. SVSPs, in particular, are multifunctional enzymes that interfere with several physiological functions of the prey or victims, such as blood clotting, fibrinolysis, blood pressure, and platelet aggregation. Some SVSPs that mimic the action of human thrombin are called thrombin-like enzymes (SVTLEs) and are present in several snake venoms [9,10]. The SVTLEs present in viperids act on fibrinogen, but only release the fibrinopeptide A [11,12].

According to the WHO, serum therapy is essential in cases of snakebites [13]. The pentavalent, bothropic antivenom (BAV) produced at the Butantan Institute is a serum obtained using venoms from *Bothrops jararaca* (50%), *B. jararacussu* (12.5%), *B. alternatus* (12.5%) *B. moojeni* (12.5%), and complex *B. neuwiedi* (12.5%) [14,15]. While *Bothrops atrox* venom is not specifically included in the antigen pool used for the manufacture of BAV, it is utilized in cases of accidents caused by this species and all snakes of the *Bothrops* genus found in Brazil, due to the similarities in the composition of their venoms and the symptoms presented by the victims [16]. Even though BAV has saved countless lives [16,17], some studies have shown failures in its ability to neutralize certain components present in bothropic venoms, especially SVSPs [18,19,20,21], underscoring the need for its improvement. Furthermore, the development of new antivenoms is important and, in this scenario, monoclonal antibodies (mAbs) are raised as an alternative and promising tool to the currently used serum therapy [22]. Antibody affinities and specificities crucially depend on six hypervariable loops at the antigen-binding site, known as complementarity-determining regions (CDRs). The portions called CDR3 are most diverse in their length and composition and therefore are considered the most decisive for antigen recognition. Therefore, this study uses small synthetic peptides homologous to the CDR3 regions of mAb 6AD2-G5 as inhibitors of SVSPs present in bothropic venoms.

## 2. Results

### 2.1. Purification of the Anti-Thrombin-like Enzyme Antibody (6AD2-G5)

The priority of this study was to verify the modulatory capacity of synthetic peptides homologous to the CDR3 regions of an mAb against an SVTLE from *B. atrox*. Therefore, after culturing the hybridoma, the mAb was obtained and purified. The efficient purification of the anti-thrombin-like enzyme antibody (6AD2-G5) from the culture medium was achieved. Approximately 5.0 mL of supernatant was obtained, with a yield of 0.2 mg/mL. Protein bands corresponding to the mAb, above 140 kDa in non-reducing conditions, and heavy chains ~50 kDa and light chains ~25 kDa in reducing conditions, can be observed in Panels A1 and A2, respectively. Also, they were recognized in Western blot analyses using the *B. atrox* venom as the antigen (Panels B1 and B2, non-reducing and reducing conditions, respectively) (Figure 1). The high-molecular-mass bands observed in Panel A1 (above 260 kDa) could be possible complexes between the mAb molecules. The antibody showed the same characteristics as previous publications by Petretski et al., 2000, and Frauches et al., 2013 [23,24].

### 2.2. Purification of the SVTLE

Purified SVTLE from *B. atrox* venom was obtained according to the methodology described by Petretski et al., 2000 [23], with small modifications. Initially, the venom was fractionated on a gel-filtration column (Appendix A) and the obtained fractions were analyzed by SDS-PAGE. As shown in Figure 2, lanes 6, 7, and 8 present a protein band of approximately 42 kDa, which is the expected molecular mass for the SVTLE of *B. atrox* venom. Moreover, samples 6, 7, and 8 showed specific activity on the substrate FRET Abz-Serine, selective for SVSPs of bothropic venoms [19] of 95 UF/min/µg, 67 UF/min/µg, and 36 UF/min/µg, respectively. In addition to the 42 kDa protein band, it is also possible to observe other components present in fractions 6, 7, and 8 with molecular masses around 40 kDa and 22 kDa, which are probably metalloproteases of the P-II and P-I classes, respectively [25].

Continuing with the purification of SVTLEs, fractions 6–8 were pooled and applied to an anion exchange column (Figure 3). The resulting fractions were analyzed by SDS-PAGE (12%), and fractions 35 and 36 showed protein bands with molecular weights around 42 kDa and hydrolytic activity on the Abz-Ser substrate (220 UF/min/µg).

### 2.3. Monoclonal Antibody Digestion and Mass Spectrometry Analyses

The bands corresponding to the light and heavy chains of the antibody, on SDS-PAGE under reducing conditions (Figure 1), were digested separately with trypsin and chymotrypsin, as described in the Methods section. The hybridoma-derived antibody sequences obtained from the approaches PEAKS DB, PEAKS SPIDER, and de novo, a hybrid PSM from the LC-MS/MS experiment using heavy and light chains, resulting in 467,288 sequences of 3–73 amino acids in length (Table 1). Despite the relatively short size of the sequences, they were useful for our de novo assembly approach. The rates of somatic hypermutation are generally low (1–10%) and the translated germline sequences contained in the IMGT may have sufficient homology to correctly identify the peptide sequences and to be located in the correct framework region.

Based on this, we developed our two-step approach, starting with the assembly of antibody-derived de novo sequences using the PASS and ALPS assemblers [26], and further, the extender approach, using in-house scripts and the fasta36 [27] tool to perform template-based antibody assembled sequence extension. The assembly resulted in 2741 sequences of 6–380 amino acids in length. The approach can be performed on any user-defined set of templates, using plain FASTA sequences as input, and pre-defining a set of frameworks as a template in FASTA format.

The peptide sequences obtained were RDWDGYYFDY, ARLPDDHYFDY (from the VH domain), and QSYNLRT (from the VL domain) (Figure 4).

After sequencing the CDR3 regions, some insertions of amino acids at the N-terminal in the three peptides were made. In addition to being observed at the C-termini of framework 3 (Figure 4a), cysteine residues are considered important for the maintenance of CDR3 loops [28,29,30]. In addition, positively charged arginine or lysine can form a salt bridge with a conserved aspartate encoded by the JH region, thus stabilizing the base of the CDR loop [31,32,33].

The first synthesized peptides were CKRDWDGYYFDY, CARLPDDHYFDY, and CKQSYNLRT. After results were obtained with this first generation of peptides, and based on the Ki values determined with the SVTLE, two of these peptides were linked by glycine and proline sequences, and two new peptides were synthesized: CKRDWDGYYFDYGGPGCKQSYNLRT and CKRDWDGYYFDYGGPGGCKQSYNLRT. The amino acids proline and glycine were used as linker sequences due to their ability to form flexible structures and avoid steric interference between adjacent functional domains [34,35].

### 2.4. Determining the Blocking Activity of SVSPs by Synthetic Peptides

The synthetic peptides CKRDWDGYYFDY, CARLPDDHYFDY, CKQSYNLRT, CKRDWDGYYFDYGGPGCKQSYNLRT, and CKRDWDGYYFDYGGPGGCKQSYNLRT were named CDR3 A to CDR3 E, respectively. The substrate Abz-Ser, characterized as selective for SVSPs present in bothropic venoms [19], and the venoms of *B. jararaca* and *B. atrox* were used.

The results indicate that four peptides—CDR3 A, CDR3 C, CDR3 D, and CDR3 E—have effective inhibitory activity of Abz-Ser hydrolysis, as can be seen in Figure 5. As an exception, the peptide called CDR3 B did not behave as an inhibitor of the SVSPs present in both studied venoms. The new-generation peptides, CDR3 D and CDR3 E, proved to be effective regarding the inhibitory potential of SVSPs present in both studied venoms. CDR3 D showed inhibition of Abz-Ser hydrolysis of 57% and 50% for SVSPs present in *B. jararaca* and *B. atrox* venoms, while CDR3 E showed an inhibitory capacity of 47% for both venoms (Figure 5).

### 2.5. Comparison of the Blocking Potential of BAV and Synthetic Peptides

For comparison purposes, we conducted serum neutralization assays using bothropic antivenom and peptides homologous to the CDR3 regions of the mAb 6AD2-G5. The commercial antibothropic serum produced by the Butantan Institute was able to partially inhibit the hydrolysis of the FRET Abz-Ser substrate by the *B. atrox* venom when used in a ratio of 1:100 (venom/antivenom), as shown in Figure 6. On the other hand, in assays with the *B. jararaca* venom, BAV was not able to inhibit the hydrolysis of the used substrate, even at higher doses (1:500, venom/antivenom). These results agree with the literature, since this failure of BAV to inhibit SVSPs present in Bothrops snake venoms has already been previously reported [19,20,21]. The four synthetic peptides that showed inhibition of Abz-Ser hydrolysis (CDR3 A, CDR3 C, CDR3 D and CDR3 E, Figure 5) showed superior efficacy when compared to BAV, being more effective in the assays with the *B. jararaca* venom. CDR3 E (10 µM) was more efficient for inhibiting Abz-Ser hydrolysis when compared to BAV (Figure 6).

### 2.6. Susceptibility of Synthetic Peptides to Hydrolysis by Bothrops jararaca and B. atrox Venoms

To verify the stability of the synthetic peptides, they were incubated with both venoms, and tests were performed using reverse-phase liquid chromatography on a C-18 column. As shown in Table 2, and in Appendix A, the synthetic peptides were resistant to hydrolysis by both *B. jararaca* and *B. atrox* venoms even after 4 h of incubation at 37 °C. As a positive control, the β-chain of insulin was used, which is known to be a substrate for the SVSPs and SVMPs of bothropic venoms [36]. Since the peptides did not behave as substrates for the venoms, they can be considered inhibitors (peptides CDR3 A, CDR3 C, CDR3 D, and CDR3 E) of SVSPs.

### 2.7. Studies with the SVTLE of Bothrops atrox—Determination of the Mechanisms and Inhibition Constants (Ki) of Peptides

To obtain the inhibition constants (Kis) of the peptides for the hydrolysis of Abz-Ser by the SVTLE purified from *B. atrox* venom, it was necessary, initially, to evaluate the Michaelis–Menten constant (Km). For that, five substrate concentrations (1.0 µM, 2.5 µM, 5 µM, 7.5 µM, and 10 µM) and 5.0 ng of the SVTLE were used. The Km value was determined to be 18.80 µM ± 2.4, and the maximum velocity (Vmax) attained was 1579.0 ± 2.4 UF/min/µM (Appendix A). The substrate was efficiently hydrolyzed by the SVTLE, presenting a specificity constant of 84 µM^−1^ UF/min^−1^.

To define the mechanism of action of the peptides, we utilized two concentrations of Abz-Ser and two concentrations of the synthetic peptides. For each substrate concentration, a control test in the absence of the peptide was performed. The assays were performed in duplicate, and during all experiments, the consumption of substrate was maintained under 10%. The reactions were monitored in the fluorimeter, as described above, and were analyzed in Grafit 5 software (Erithacus Software, West Sussex, UK). The mechanism and the inhibition constants were determined through the Dixon plot (1/V × [I]) equation [37]. The Dixon plot (1/V versus [I]) (Figure 7) shows that the four peptides studied are non-competitive SVTLE inhibitors, and the Ki values obtained were 1.06 µM (CDR3 A), 3.24 µM (CDR3 C), 0.48 µM (CDR3 D), and 0.60 µM (CDR3 E).

### 2.8. Analysis of the Inhibition Selectivity of Synthetic Peptides

As shown in Figure 8, CDR3 A reduced the hydrolytic activity of thrombin by 34%, while CDR3 E showed inhibition of the substrate hydrolysis of Abz-FRSSRQ-EDDnp both in studies with trypsin (24%) and thrombin (100%). In the analysis with elastase-1, only CDR3 C was able to reduce the hydrolysis of the substrate used by 22%.

To elucidate the inhibition mechanism of the two synthetic peptides, which exhibited inhibition of human thrombin activity, fluorometric tests were performed using three different concentrations of each peptide (including a positive control, with the inhibitor concentration being equal to zero). With the obtained velocities, curves were plotted on the Dixon plot (1/V versus [I]) (Figure 9). After the mechanisms of inhibition were verified, the inhibition constant (Ki) was determined as 61.8 µM (CDR3 A) and 55.8 µM (CDR3 E).

In addition to studies with serine proteases, assays on the inhibition of metalloproteases present in *B. atrox* and *B. jararaca* venoms were carried out. For this, the FRET substrate Abz-FASSAQ-EDDnp (Abz-Metal) was used, since it is selective for metalloproteases from bothropic venoms [19]. The results indicated that CDR3 E was capable of inhibiting the hydrolysis of Abz-Metal by 30%, when using 100 µM of the peptide, in studies with *B. jararaca* venom. Regarding the results obtained with the *B. atrox* venom, only CDR3 C was able to reduce substrate hydrolysis by 10%, also when used at a concentration of 100 µM (Figure 10).

### 2.9. In Silico Analysis for the Prediction of Hemolytic Activity

The possible hemolytic activity of the peptides was analyzed through in silico studies. The results obtained on the HemoPI platform: Hemolytic Peptide Identification Server (https://webs.iiitd.edu.in/raghava/hemopi/, accessed on 29 November 2023) are shown in the Table 3, indicating that the peptides do not present hemolytic activity.

## 3. Discussion

As previously noted, the majority of snakebites in the Brazil area are attributed to snakes of the *Bothrops* genus, mainly *B. jararaca* in the South and Southeast regions and *B. atrox* in the Brazilian Amazon [5,6]. For this reason, although the synthetic peptides are homologous to the CDR3 regions of a mAb against an SVTLE present in *B. atrox* venom, investigations involving *B. jararaca* venom were carried out. It is worth highlighting that the bothropic antivenom (BAV) produced by the Butantan Institute is obtained from the use of five bothropic venoms as antigens and is administrated to treat bites from all snakes of this genus found in Brazilian territory, totaling 23 species. Considering that venoms have intraspecific and interspecific variability [15,16], the development of an antibothropic serum of national coverage is a challenge.

The BAV demonstrates efficacy in neutralizing the systemic effects caused by bothropic venoms and remains the sole treatment recommended by the WHO [13]. However, findings from previous studies conducted by our group suggest a shortfall in BAV’s capacity to neutralize bothropic SVSPs, signaling the need for improvements in the treatment of snakebites. To elucidate this, our investigations reveal that at least three SVSPs present in *B. jararaca* venom evade neutralization by BAV, despite being immunogenic. Specifically, although the antibodies within BAV recognize these serine proteases in ELISA and Western blot assays, they are not inhibitors of their proteolytic activity [21]. In addition, local damage is not well neutralized by conventional therapy and may result in permanent disability and amputation of the affected limb. This occurs mainly due to the rapid action of some toxins, making treatment even more challenging, especially when snakebites occur in hard-to-access regions of the country [4,14,38,39]. Another negative point of therapy with current antivenoms is that their production is a complex process of obtaining the antigens that involves the maintenance of snakes and extraction of the venom, the maintenance and immunization of large mammals, and obtaining and purifying the plasma to obtain the antigens. It is also important to highlight that current antivenoms can cause undesirable side effects, such as serum sickness [40,41].

Given the aforementioned challenges, the search for alternatives that help in the treatment of snake envenomation has strongly motivated the scientific community.

Snake venoms are a complex mixture of different toxins that have a wide range of physiological effects; among them are SVSPs [7,8]. Serine proteases from snake venoms are classified in the PA clan, chymotrypsin family S1, and exhibit the highly conserved catalytic triad (His43, Asp88, and Ser184, chymotrypsin numbering) [42]. Unlike trypsin, SVSPs are characterized by a high specificity for the hydrolysis of their substrates, despite having a high degree of identity among their primary sequences. Typically, SVSPs exhibit approximately 51–98% identity with each other, 26–33% with human thrombin, and 34- 40% with human plasma kallikrein [43]. In general, SVSPs affect the coagulation cascade by activating components involved in coagulation, fibrinolysis processes, platelet aggregation, and hypotension through mechanisms that mimic mammalian enzymes. Due to the fact that some SVSPs mimic the action of thrombin, they have been named thrombin-like enzymes (TLEs) [43].

The significance of SVSPs as key toxins that participate in envenoming and the ineffectiveness of BAV in neutralizing the activities of these enzymes underscore the need for the development of next-generation antivenoms and were the motivation of the present work. Thus, *B. atrox* and *B. jararaca* venoms, together with a purified SVTLE from *B. atrox* venom, were used to analyze the inhibitory capacity of five synthetic peptides homologous to the CDR3 regions of a monoclonal antibody produced by the hybridoma 6AD2-G5.0. Enzymatic activity assays with the peptides were performed using the substrate Abz-Ser, which was previously characterized by our group as selective for bothropic venom serine proteases [19].

It was possible to observe the ability of four synthetic peptides homologous to the CDR3 regions of the mAb (CDR3 A, CDR3 C, CDR3 D, and CDR3 E) to inhibit the SVSPs present in *B. jararaca* and *B. atrox* venoms, in addition to the purified SVTLE. It is important to mention that two sequences corresponding to possible CDR3 portions originating from the VH domain were sequenced by mass spectrometry analyses. Thus, in the uncertainty of the correct sequence, the two peptides were synthesized and named CDR3 A and CDR3 B. Since CDR3 B did not show interactions with either venom studied, as well as the purified SVTLE, CDR3 A must correspond to the sequence present in the mAb.

The comparative analysis between the inhibitory effect of the peptides and BAV indicated that the SVSPs present in *B. jararaca* venom are more effectively inhibited by synthetic CDR3 regions. On the other hand, both synthetic peptides and BAV equivalently inhibit the SVSPs of *B. atrox* venom. These studies also allowed the observation of the intraspecific variation of bothropic venoms, as BAV was more effective in neutralizing the SVSPs of a venom not used for its manufacture, in this case, the venom of *B. atrox*, than the same class of enzymes of *B. jararaca* venom (whose venom makes up 50% of the immunization pool). Although unexpected, our results are in accordance with the literature, where studies involving several antivenoms have demonstrated that it is not necessary to include certain species in the immunization pools used for antivenom production [44].

Despite the results obtained with CDR3 B, the other four peptides studied behaved as inhibitors of Abz-Ser hydrolysis by venoms and by the SVTLE. Kinetic studies with the purified enzyme demonstrated that all are non-competitive inhibitors of the enzyme and, consequently, indicated that they do not interact with the active site of the SVTLE. Thus, an enzymatic reaction regulated by a non-competitive inhibitor leads to less efficient catalysis, regardless of the amount of substrate available, and, therefore, higher concentrations of a non-competitive inhibitor should achieve higher inhibition levels. The most effective inhibitor, with a Ki of 0.66 μM, was CDR3 E, which was designed from CDR3 A and CDR3 C, linked by a stretch of glycine and proline residues (CKRDWDGYYFDYGGPGGCKQSYNLRT). A relevant result was the observation that the four peptides with inhibitory action were more or equally effective in blocking Abz-Ser hydrolysis than BAV. In addition to inhibiting the SVTLE present in the venom of *B. atrox*, the peptides CDR3 A and CDR3 E also proved to be inhibitors of human thrombin; however, the values of the inhibition constants obtained were about 60 times higher. Furthermore, thrombin inhibition by CDR3 A and CDR3 E presented a competitive inhibition mechanism. Although unexpected, this result can be considered beneficial, as competitive inhibitors are reversible and the propensity for bleeding should be less problematic in humans.

Metalloproteases are considered important and abundant toxins present in bothropic venoms and, therefore, experiments on their possible interaction with the synthetic CDR3 peptides were carried out. Only CDR3 E was able to inhibit the metalloproteases present in *B. jararaca* venom when used at a concentration 20 times higher than that used in studies with serine proteases. This is an unexpected result, but the simultaneous inhibition of toxins belonging to the metalloprotease and serine protease classes could be of clinical interest. Although the other serine proteases studied were not inhibited by the synthetic peptides, a broader panel of proteases, in addition to serine proteases, should be studied to verify the selective inhibition of SVSPs.

Since the production of the first antivenom by Hawgood and Calmette in 1999, obtained by inoculating rabbits with the venom of the *Naja tripudians* snake [45], few changes have been made, such as the purification of IgGs by caprylic acid and the use of papain and pepsin to obtain F(ab’)2 or F(ab) fragments, respectively [46]. Therefore, it is extremely important to develop new pharmacological tools to support and improve the currently used immunotherapy, given the need to improve the neutralization of toxins present in animal venoms [46]. Likewise, challenges exist in the production of antivenoms, as well as difficulties in distribution, and the requirements for specialized personnel to administer antivenoms, among other factors.

The inhibitory potential demonstrated by the synthetic peptides of both SVSPs present in total venoms and the SVTLE of *B. atrox* is promising. Indeed, there is a need for the design of new sequences, aiming to create a mimetic peptide that exhibits potency, selectivity, and stability, for use as support in immunotherapy. In the future, it is expected that the association of molecules with inhibitory potential against the main classes of toxins present in bothropic venoms will emerge as an effective alternative for the treatment of ophidian accidents, without the need to use serum-based antivenoms.

## 4. Materials and Methods

### 4.1. Reagents

Trifluoroacetic acid (TFA) and acetonitrile (ACN) were obtained from Sigma-Aldrich. The Fluorescence Resonance Energy Transfer (FRET) substrates, Abz-Ser (Abz-RPPGFSPFRQ-EDDnp) and Abz-FRSSRQ-EDDnp, synthesized by the solid-phase synthesis method, were kindly provided by Prof. Dr. Luiz Juliano Neto from the Biophysics Department of UNIFESP/EPM. Peptides homologous to CDR 3 portions were obtained by the solid-phase peptide synthesis method (F-moc) and purchased from GenOne Biotechnologies, with a purity greater than 95% (Rio de Janeiro, Brazil).

### 4.2. Venoms and Antivenom

The venoms of *B. atrox* and *B. jararaca* (50 mg) were provided by the Butantan Institute Venom Commission, São Paulo, Brazil. Venom stock solutions (1.0 mg/mL) were prepared in PBS buffer, containing 50 mM sodium phosphate and 20 mM NaCl, pH 7.4. Bothropic antivenom (BAV) was supplied by the Hyperimmune Plasma Processing Section of the Butantan Institute, São Paulo, Brazil (Lots 0805063 and 056110). Experimental procedures were approved by the Institutional Committee for the Care and Use of Laboratory Animals from Butantan Institute (CEUAIB protocol number 5461 160523).

### 4.3. Hybridoma 6AD2-G5 Culture and Purification

The hybridoma, 2 × 10^6^ viable cells/mL, was cultured (CELLine Bioreactor Flasks–CL350) in 20.0 mL DMEM/F12 medium (Gibco, Invitrogen Corp, Carlsbad, CA, USA), supplemented with 10% fetal bovine serum and 10 µg/mL gentamicin (Gibco, Invitrogen Corp, Carlsbad, CA, USA). Every 7 days, the medium in the cell compartment of the bioreactor, containing the 6AD2-G5 antibody, was collected by centrifugation at 400 g for 5 min. Cells were then diluted, and 2 × 10^6^ viable cells/mL (20.0 mL of final volume) were returned to the bioreactor cell compartment for continued antibody production. At each collection, an aliquot of cells was stored at −80 °C in TRIzol for subsequent RNA extraction.

Three milliliters of the cellular compartment medium was fractionated by salting out with ammonium sulfate, dialyzed on a 10 kDa cut-off membrane, and the antibody was purified using affinity chromatography on a HiTrap Protein A column (GE Healthcare, Chicago, IL, USA). The equilibrium and elution of unretained components were monitored by absorbance measurements at 280 nm (Hidex Sense 425-301 microplate reader, Turku, Finland), using 20 mM sodium phosphate buffer, pH 7.0 (approximately 20 mL). Afterward, the mAb was eluted with the elution buffer (100 mM citric acid pH 3, approximately 5 mL). In the vial containing the eluted antibodies, 100 µL of 1 M Tris-HCl pH 9 solution was previously added to maintain the stability of the mAb. Protein concentration was determined using a BCA Protein Assay Kit (Pierce Biotechnology, Rockford, IL, USA)) and serum albumin (BSA, Aldrich, MO, USA) as a reference in a Hidex Sense microplate reader (Hidex Sense 425-301 microplate reader, Turku, Finland). Purification was evaluated using SDS-PAGE 7.5% under non-reducing conditions and subjected to silver nitrate impregnation. Proteins (5 µg) separated on SDS-PAGE were transferred to nitrocellulose membranes and evaluated by Western blot using alkaline phosphatase-conjugated goat anti-mouse IgG (1:7500).

### 4.4. Sequencing of the Anti-Thrombin-like Antibody (6AD2-G5)

The protein bands corresponding to the light and heavy chains of the antibody in SDS-PAGE under reducing conditions were digested separately with trypsin and chymotrypsin, according to Shevchenko et al., 2006 [47], with a modification for digestion with chymotrypsin. With this enzyme, the enzymatic reaction proceeded for 48 h at an enzyme concentration of 40 ng/µL. Peptides from the enzymatic digestion were then desalted, concentrated, and resuspended in 0.1% formic acid. Mass spectrometric analysis was performed by online liquid chromatography in a nano-flow chromatograph (Easy-nLC Proxeon) coupled to a Velos LTQ-Orbitrap Velos spectrometer (Thermo Fisher Scientific, Bremen, Germany) and a 10 cm column (75 µm i.d. × 350 µm e.d.) packed with 5 µM Jupiter C-18 beads (Phenomenex, Torrance, CA, USA). For data analysis, the raw files (.RAW) were subjected to searches using PEAKS Studio software (version 8.5) [48] against the immunoglobulin database made available by Tran et al., 2016 [49], as well as the IMGT (International Immunogenetics Information System) database [50]. As a control, the cRAP contaminant database and the MaxQuant contaminant database were used. For variable region sequencing, the parameters used were peptides found in both trypsin and chymotrypsin digestions, Average Local (ALC) scores of 50%, 10 ppm error for the precursor ion mass, 0.02 Da tolerance for precursor ion fragments, and several spectra observed for each peptide, at least 10.

A complete MS scan was acquired in the *m*/*z* range of 300–1650 followed by MS/MS acquisition using collision-induced dissociation (CID) of the ten most intense ions from the MS scan using an isolation window width of 3 *m*/*z*. MS spectra were acquired in the Orbitrap analyzer (scan range: 400 *m*/*z*–2000 *m*/*z*; full scan resolution: 60,000; scan resolution of MS/MS: 7500). Dynamic exclusion was defined by a list size of 500 features and an exclusion duration of 90 s. For the survey (MS) scan, a target value (AGC) of 1,000,000 and a maximum injection time of 100 ms were set, whereas the target value for the fragment ion (MS/MS) spectra was set to 40,000 ions with a maximum injection time of 100 ms. The lower threshold to target precursor ions in the MS scans was 500 counts per scan. The 24 resulting raw data files, 11 for chymotrypsin (7 for the light chain and 4 for the heavy chain) and 13 for trypsin (8 for the light chain and 5 for the heavy chain), were used for data analysis.

The raw data were imported into PEAKS Studio 8.5, pre-processed (precursor mass correction, MS/MS deisotoping and deconvolution, peptide feature detection, removal of identical peptides) against the immunoglobulin database from Tran et. al., 2016 [49], and the database created with Frameworks and CDR sequences from IMGT [50,51], corresponding to 77,422 sequences organized into 8861 FR1, 9296 FR2, 9463 FR3, 8331 FR4, 9156 CDR1, 9314 CDR2, and 9292 CDR3 from heavy regions and 2012 FR1, 2080 FR2, 2097 FR3, 1353 FR4, 2036 CDR1, 2078 CDR2, and 2053 CDR3 from light regions from *Mus musculus* and *Homo sapiens* species.

PEAKS de novo sequencing was performed with precursor and fragment error tolerances of 10 ppm and 0.02 Da, respectively. Carbamidomethylation (Cys) was set as a fixed modification, and oxidation (Met) and deamidation (Asn/Gln) were set as variable modifications. At most, three variable modifications per peptide were allowed. The peptide sequences identified by the de novo sequencing analysis were exported along with their feature areas and positional confidence scores. The database search module in PEAKS Studio 8.5 was then used in the second stage to identify peptide spectrum matches (PSMs) against the antibody protein database from Tran et al., 2016 [49]; it also included a contaminant database from the cRAP contaminant database [52]. Other search parameters were kept the same as those used in the respective de novo sequencing analysis. Antibody sequences from specific samples commonly contain slightly different sequences than those recorded in the IMGT database. To reconstruct a true sequence, minimize the sum of de novo errors and homology mutations, and detect amino acid variants, the data sets were searched using the PEAKS SPIDER tool against an in-house-created antibody database from IMGT.

Based on the PEAKS DB, SPIDER, and de novo results, a hybrid PSM set was generated for the antibody sequencing assembly according to the following: (1) the scores of the PSMs identified by PEAKS DB must be higher than FDR 1.0%; (2) the PSMs mapped to contaminant proteins must be filtered out; and (3) the Average Local Confidence (ALC) scores of PSMs identified from PEAKS de novo sequencing must be higher than 50. The PSMs were exported with their feature areas and positional confidence scores for the subsequent assembly. The three lists of peptides together with their intensities (feature areas) and positional confidence scores were obtained from PEAKS as described in the previous procedures. Subsequently, we applied two assembler approaches using all previously generated peptides to obtain the longest peptides, the ALPS assembler [26] and the PASS Assembler [27]. The assemblers were ALPS, an assembler based on the Bruijin graph using kmer 6, and PASS, for a de novo assembly for non-target seeds based on a Perl script using default parameters with the exception of ”-m 3 -o 1 -r 0.51”. The final assembled protein sequences (contigs) produced by each assembler are combined into the final data set. Finally, a set of protein sequences from frameworks 3 and 4 of interest were aligned using fasta36 with an e-value of 0.5 [27] against the reference contigs to evaluate the matched contigs, determine their relative positions and overlapping regions, and extend the contig regions, identifying new CDR3 regions.

### 4.5. Purification of a Thrombin-like Enzyme Present in the Bothrops atrox Venom

A purified SVTLE from *B. atrox* venom was obtained according to the methodology described by Petretski et al., 2000 [23], with small modifications. Initially, the venom was fractionated in an ÄKTA purifier system (Pharmacia Biotech AB, Amersham, ENG) using a Superdex 75 10/300 GE Healthcare gel-filtration column. The fractions were analyzed by SDS-PAGE (12%) under non-reducing conditions, and also tested with a fluorometric assay (Hidex Sense microplate reader; 425-301 Hidex, Turku, Finland), with excitation and emission fluorescence set at 320 nm and 420 nm, respectively, using the fluorescent substrate Abz-Ser (5 µM) in 100 mM Tris-HCl, 20 mM NaCl, pH 7.4 (final volume of 100 µL), in a 96-well plate (Corning). Those that presented bands with an MW around 42 kDa and enzymatic activity against Abz-Ser were grouped. Continuing with the purification of the SVTLE, grouped fractions were applied to an anion exchange column (HiTrap DEAE FF) previously equilibrated with 0.02 M pH 5.6 Na-acetate buffer, and the bound protein fractions were eluted with a buffer, which additionally contained 1 M NaCl using a gradient from 0 to 100% NaCl. Fractions of 1 mL/tube were collected, and the absorbance was monitored at 280 nm. The fractions (3 µg) were evaluated by SDS-PAGE (12%) under non-reducing conditions, and those that presented single bands with MWs of 42 kDa had their catalytic activities analyzed with the Abz-Ser substrate, as described above. The active fractions were pooled and frozen until use.

### 4.6. Enzymatic Assays with B. atrox and B. jararaca Venoms

The evaluation of the enzymatic activity was performed in 100 mM Tris-HCl, 20 mM NaCl, pH 7.4 (final volume of 100 µL), using a 96-well plate (Corning) and the substrate Abz-Ser (5 µM). All reactions occurred at 37 °C and, aiming for a substrate hydrolysis limit of 10% (initial hydrolysis rates), concentrations of 0.09 µg/µL *B. jararaca* venom (90 ng) and 0.02 µg/µL *B. atrox* (20 ng) were used. Assays were performed in a Hidex Sense microplate reader (425-301 Hidex, Turku, Finland), with excitation and emission fluorescence set at 320 and 420 nm, respectively. Measurements of peptidase activity were taken for 15 min continuously (one read per minute). The fluorometric assays were analyzed using Grafit 5.0 from Erithacus Software (version 5.0.6, 1989–2003, Erithacus Software, West Sussex, UK), and the hydrolysis rates (UF/min) were determined. All fluorometric measurements were made in quadruplicate, and the results are shown as means with SDs. Protein concentration was determined using the BCA Protein Assay Kit (Pierce Biotechnology, Rockford, IL, USA) and serum albumin (BSA, Aldrich, MO, USA) as a reference in a Hidex Sense microplate reader (425-301 Hidex, Turku, Finland).

### 4.7. Serum Neutralization

The neutralization capacity of the BAV against the proteolytic activity of *Bothrops atrox* (20 ng) and *B. jararaca* (90 ng) venoms was measured using the substrate Abz-Ser (Abz-RPPGFSPFRQ-EDDnp). Serum neutralization assays were performed with a 30 min room temperature pre-incubation of the venoms together with the BAV in 100 mM Tris-HCl, 20 mM NaCl, pH 7.4 buffer, and then, the substrate Abz-Ser was added. The substrate concentration was kept at 5 µM and BAV concentrations were used according to the method by Kuniyoshi et al., 2012 [19], aiming for the maximum inhibition of Abz-Ser hydrolysis, using antivenom/venom mass ratios of 500:1 for *B. jararaca* venom and 100:1 for *B. atrox.* All experiments were performed in quadruplicate.

### 4.8. Characterization of Synthetic Peptides as Thrombin-like Inhibitors

To obtain the inhibition constant (Ki) of the peptides on the purified thrombin-like enzyme, it was necessary to obtain the Michaelis–Menten constant (Km) for the hydrolysis of the substrate Abz-Ser by the serine protease. Four concentrations of Abz-Ser substrate (2.5 µM, 5.0 µM, 7.5 µM, and 10 µM) and one concentration of an SVTLE (20 ng) were used in 100 mM Tris-HCl, 20 mM NaCl, pH 7.4 buffer (final volume 100 µL). The reactions were continuously monitored (fluorescence adjusted to 320 and 420 nm) in a fluorimeter (425-301 Hidex, Turku, Finland), as previously described. The maximum velocity (Vmax) and Km values were obtained in Grafit 5 version software (Erithacus software, East Grinstead, West Sussex, UK). All experiments were performed in quadruplicates. Protein concentration was determined using the BCA Protein Assay Kit (Pierce Biotechnology, Rockford, IL, USA) and serum albumin (BSA, Aldrich, MO, USA) as a reference in a Hidex Sense microplate reader (425-301 Hidex, Turku, Finland).

To determine the values of the inhibition constants (Kis) and the inhibition mechanisms of the synthetic peptides on the thrombin-like activity, fluorometric assays were performed using three different concentrations of each one of the inhibitors (including a positive control, with the inhibitor concentration equal to zero, 100 µM, and 200 µM). With the velocities obtained, a Dixon plot (1/V versus [I]) was constructed. All experiments were performed in quadruplicates, with a substrate consumption of less than 10%, using concentrations of 7.5 µM and 10 µM of the substrate Abz-Ser. The results were analyzed in Grafit 5 version software (Erithacus software, East Grinstead, West Sussex, UK).

### 4.9. Stability Test of Synthetic Peptides against Venoms

With the aim of verifying the possible hydrolysis of peptides by the studied venoms, the samples were prepared in 100 mM Tris-HCl, 20 mM NaCl, pH 7.4, final volume 100 µL, using 50 µM of each synthetic peptide, incubating them separately with the venoms of *B*. *jararaca* (90 ng) and *B. atrox* (20 ng) for 4 h at 37 °C, the same concentrations used in the enzymatic assays. A solution containing the β-chain of insulin (50 µM) was used as a positive control since it is a described substrate for SVSP from both venoms [36]. The conditions for the HPLC (Shimadzu Prominence) analysis used 0.1% trifluoroacetic acid (TFA) in water (solvent A), and acetonitrile (ACN) and solvent A (9:1) as solvent B. The analyses occurred at a flow rate of 1 mL/min using a C-18 column (Shim-pack VP-ODS4.6 x 150 mm) and a gradient of 20–60% solvent B for 20 min. In all cases, 50 µL of each sample was injected and the absorbance was monitored at 214 nm. No precipitates were observed in the samples after the incubation period.

### 4.10. Study of the Selectivity of Synthetic Peptides for Inhibition of SVSPs

To verify the selective inhibition of the SVTLE by the peptides, it was necessary to perform experiments with other serine proteases of physiological relevance. For this, the possible inhibition of elastase-1 (EC 3.4.21.36), trypsin (EC 3.4.21.4), and human thrombin (EC 3.4.21.5) was analyzed. The evaluation of enzyme activities was performed as described above. The activities were assessed in 50 mM Tris-HCl, pH 8, for trypsin; Tris-HCl 50 mM, NaCl 50 mM, pH 8, for elastase-1; and 100 mM Tris-HCl, 20 mM NaCl, pH 7.4, for human thrombin (final volume of 100 µL), using 96-well plates (Corning) and the substrate Abz-FRSSRQ-EDDnp (5 µM). All reactions occurred at 37 °C using 1.7 ng of trypsin, 300 ng of human thrombin, and 300 ng of elastase-1. Assays were performed on a Hidex fluorimeter (425-301 Hidex, Turku, Finland), with the excitation and emission of fluorescence fixed at 320 and 420 nm, respectively. The trypsin and elastase-1 assays were performed using 20 µM of each peptide, and the human thrombin assays were performed using 100 µM. The readings occurred over 15 min at 1 min intervals. The experiments were performed in quadruplicate.

### 4.11. Evaluation of the Inhibition of Metalloproteases from Bothropic venoms by Synthetic CDR3

The assays were performed in 100 mM Tris-HCl buffer, 20 mM NaCl, pH 7.4 (final volume 100 µL), using 96-well plates (Corning) and the Abz-FASSAQ-EDDnp substrate (Abz-Metal, 5 µM). All reactions occurred at 37 ºC using a concentration of 50 ng of *B. jararaca* venom and 20 ng of *B. atrox* venom. The assays were performed on a Hidex fluorimeter (425-301 Hidex, Turku, Finland), with excitation and emission fluorescence set at 320 and 420 nm, respectively. Readings were obtained for 15 min at 1 min intervals, resulting in 15 measurements. Experiments were performed in quadruplicate.

### 4.12. Prediction of Hemolytic Peptides

The HemoPI platform: Hemolytic Peptide Identification Server (https://webs.iiitd.edu.in/raghava/hemopi/, accessed on 29 November 2023) was used to predict the hemolytic activity of synthetic peptides. Indolicidin (ILPWKWPWWPWRR), described as having broad hemolytic activity, was used as a positive control. The “SVM (HemoPI-1) based” prediction method was used for the analysis and the PROB score, ranging between 0 and 1 (where 1 indicates a high probability of being hemolytic and 0 indicates a low probability).

### 4.13. Statistical Analyses

The results obtained were statistically analyzed in the GraphPad Prism 8 program, using the one-way ANOVA test followed by Tukey’s post-test. Results with a *p*-value <0.05 (*) were considered significant. The statistical power check was carried out using the G* Power 3.1.9.7 software [53].

## 5. Conclusions

Although preliminary, the results obtained confirm the potential use of small peptides derived from complementary determinant regions as inhibitors of snake venom toxins, laying the foundation for the development of next-generation antivenoms. When the peptides presenting a putative CDR3 of the heavy and light chains were joined, the level of inhibition was similar. Thus, new peptides should be designed with the objective of obtaining a mimetic peptide that can be used in the treatment of snakebites.

## Figures and Tables

**Figure 1 ijms-25-05181-f001:**
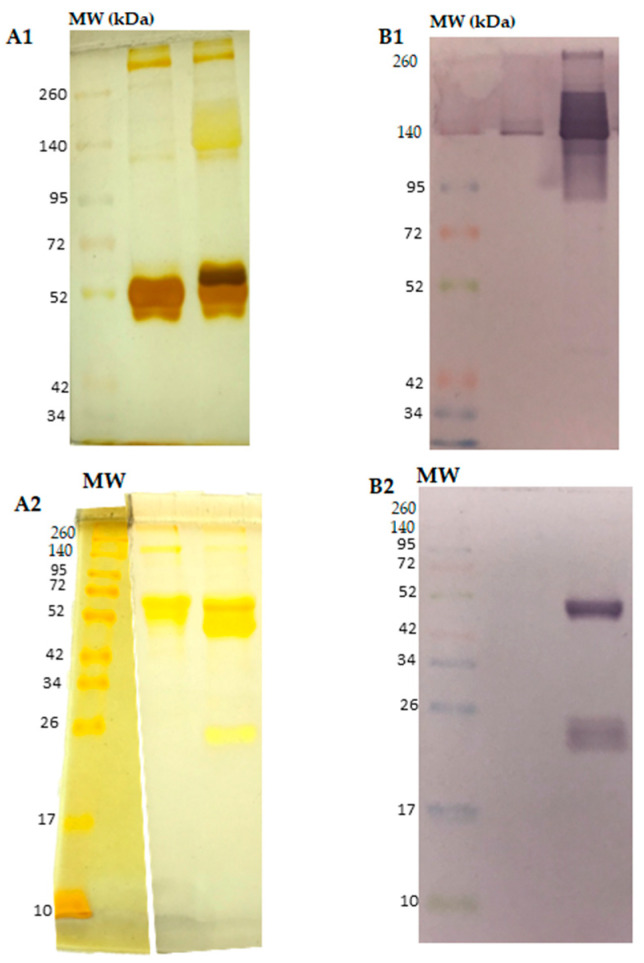
Panel (**A1,A2**): mAbs (5.0 µg) purified by salting out with ammonium sulfate and affinity chromatography on a HiTrap Protein A column (GE Healthcare) analyzed by SDS-PAGE 7.5% and visualized with silver nitrate in non-reducing and reducing conditions (Panels **A1** and **A2**, respectively). Panel (**B1,B2**): Western blot analyses of the mAbs in non-reducing and reducing conditions (Panels **B1** and **B2**, respectively). Membranes were developed with an anti-mouse antibody, 1:7500, conjugated with alkaline phosphatase. Lane 1: culture medium; lane 2: culture medium after 7 days of cultivation. Molecular weight markers (MW, ranging from 10 to 260 kDa; 5 µL/lane) were introduced into the last lanes in all runs.

**Figure 2 ijms-25-05181-f002:**
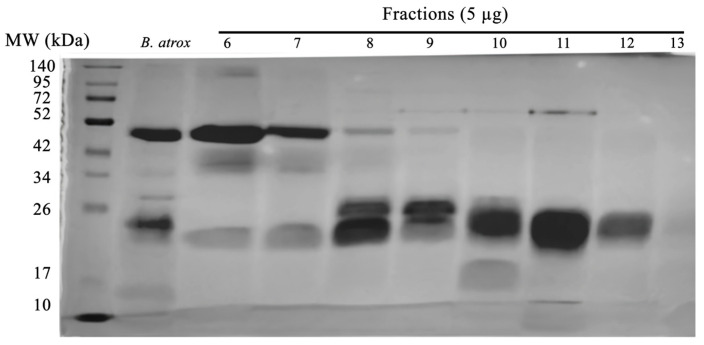
The SDS-PAGE (12%) results of the fraction eluted from the gel-filtration chromatography. Lane MW: molecular mass markers; lane *B. atrox*: *B. atrox* venom (5 µg).

**Figure 3 ijms-25-05181-f003:**
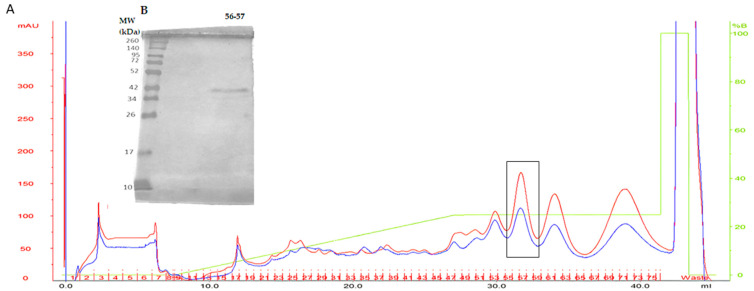
Panel (**A**): Anion exchange chromatography. Samples from tubes 6, 7, and 8 of the molecular exclusion chromatography were pooled and applied to the anion exchange column. The red and blue lines indicate the two chromatograms performed. Panel (**B**): The resulting fractions were analyzed by SDS-PAGE (12%) and fractions 56 and 57, indicated by the open box in the chromatogram, show protein bands with molecular weights around 42 kDa and hydrolytic activity on the Abz-Ser substrate (220 UF/min/µg). The single protein band below 42 kDa corresponds to the purified SVTLE. The details of the experiments are described in the Methods section.

**Figure 4 ijms-25-05181-f004:**
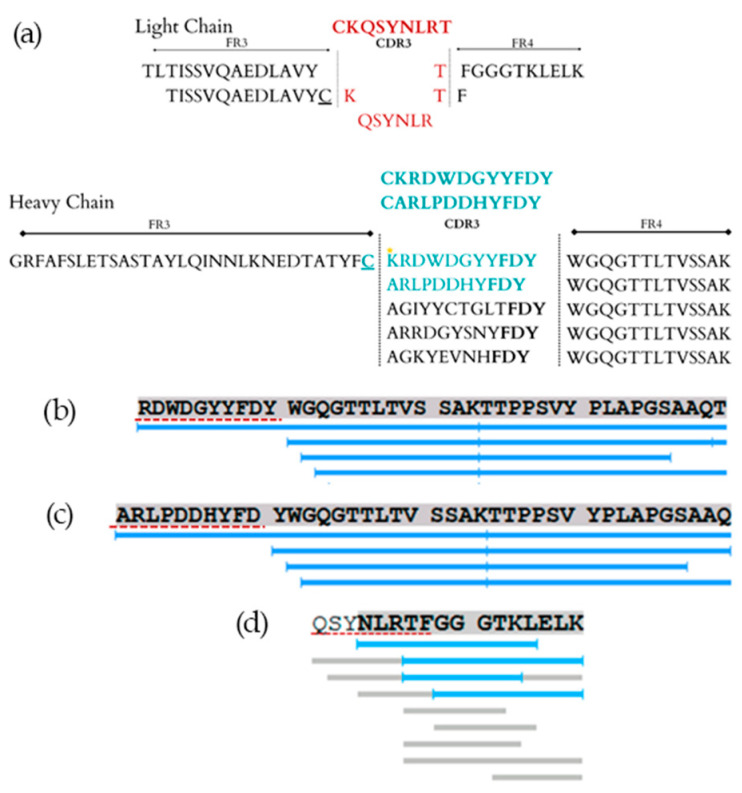
Panel (**a**) shows the sequence found in mass spectrometry flanked by framework regions, with the star indicating where the amino acid substitution was made. Panels (**b**–**d**) show the sequences obtained from the PEAKS Studio software by the de novo analysis method. (**b**,**c**) are the CDR3 sequences of the mAb heavy chain, and (**d**) shows the CDR3 from the light chain. Highlighted and underlined amino acids in blue represent the peptides sequenced by PEAKS Studio X software; the amino acids underlined in gray represent the sequences found in the de novo sequencing. The sequences obtained for the heavy chain are RDWDGYYFDY and ARLPDDHYFDY, and for the light chain, QSYNLRT, and they are indicated by the dotted underline in red.

**Figure 5 ijms-25-05181-f005:**
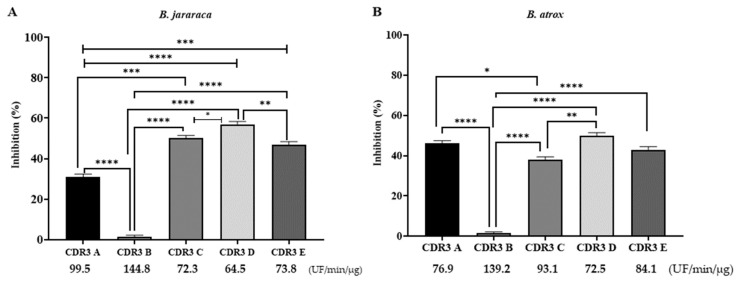
Enzyme activity assays of *B. jararaca* (**A**) and B. atrox (**B**) venoms. The *B. jararaca* (90 ng) and B. atrox (20 ng) venoms and Abz-Ser substrate (5 µM) were used in the presence of synthetic peptides. The experiments were performed in a fluorimeter (Hidex 425-301, Turku, Finland), with readings set for excitation and emission at 320 and 420 nm. Assays were performed in quadruplicate in 100 mM Tris-HCl, 20 mM NaCl, pH 7.4 at 37 °C. The peptides used at 10 µM were CDR3 A, CDR3 B, CDR3 D, and CDR3 E, and CDR3 C was used at 5 µM. The numbers below the CDRs indicate the specific activities obtained in the assays (UF/min/µg), and the specific activities obtained with the *B. jararaca* and *B. atrox* venoms were 150 UF/min/µg and 145 UF/min/µg, respectively. (*) *p* < 0.05, (**) *p* < 0.01, (***) *p* < 0.001 and (****) *p* < 0.0001.

**Figure 6 ijms-25-05181-f006:**
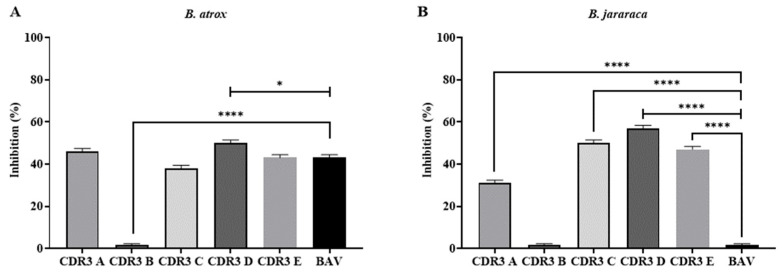
In vitro serum neutralization assays of B. atrox (**A**) and B. jararaca (**B**) venoms, performed in a fluorimeter (Hidex 425-301, Turku, Finland), with readings adjusted for excitation and emission at 320 and 420 nm. Assays were performed in 100 mM Tris-HCl, 20 mM NaCl, and pH 7.4 buffer, using the substrate Abz-Ser (5 µM). The concentrations of antivenom/venom (weight ratio of venom/antivenom) were 1:100 and 1:500 in the studies with B. atrox and B. jararaca, respectively. The synthetic peptides were used in concentrations of 5 µM for CDR3 C and 10 µM for the others (CDR3 A, CDR3 B, CDR3 D, and CDR3 E). (*) *p* < 0.05 and (****) *p* < 0.0001.

**Figure 7 ijms-25-05181-f007:**
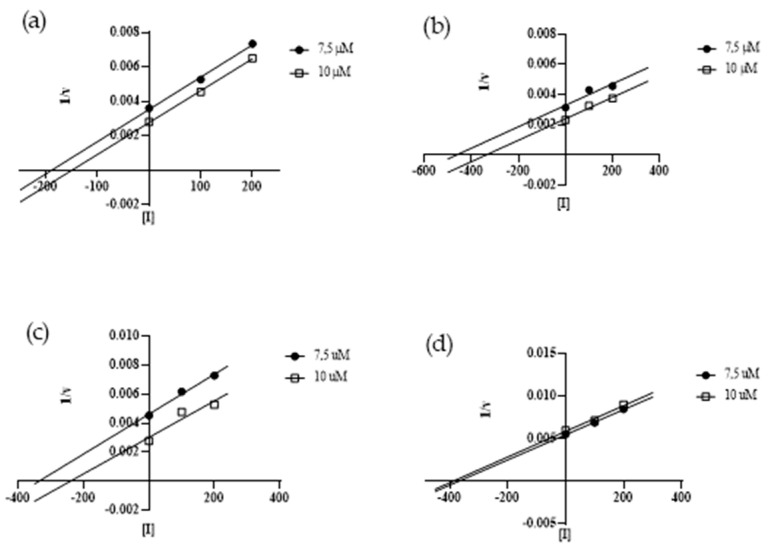
Dixon plots for the characterization of the type of inhibition exhibited by CDR3 A (panel **a**), CDR3 C (panel **b**), CDR3 D (panel **c**), and CDR3 E (panel **d**) (including a positive control, with the inhibitor concentration equal to zero) using the substrate Abz-Ser (5.0 µM, 7.5 µM, and 10 µM) in 100 mM Tris-HCl, 20 mM NaCl, pH 7.4. The parallel lines indicate a non-competitive mechanism of action. Kis of 1.06 µM (CDR3 A), 3.24 µM (CDR3 C), 0.48 µM (CDR3 D), and 0.60 µM (CDR3 E) were obtained. For the construction of the graph, the software GraphPad Prism 8 was used.

**Figure 8 ijms-25-05181-f008:**
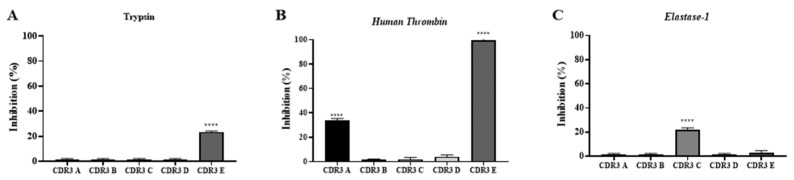
Inhibitory activity assay against human thrombin (panel **A**), trypsin (panel **B**), and elastase-1 (panel **C**). The experiments were performed in a fluorimeter (Hidex 425-301, Turku, Finland), with readings adjusted for excitation and emission at 320 and 420 nm using Abz-FRSSRQ-EDDnp as a substrate. Assays were performed in quadruplicate, in Tris-HCl 50 mM, pH 8.0 (for trypsin and elastase-1 assays), and 100 mM Tris-HCl, 20 mM NaCl, pH 7.4, for human thrombin at 37 °C, using 1.7 ng/µL of trypsin, 300 ng/µL of human thrombin, and 300 ng of elastase-1. Panel (**A**): The analyses of the peptides (20 µM) to inhibit trypsin demonstrate that only CDR3 E was able to reduce the FRET substrate hydrolysis (24%). Panel (**B**): For these assays, the peptides were used at 100 µM, and CDR3 A and CDR3 E inhibited the substrate hydrolysis by thrombin by 34% and 100%, respectively. Panel (**C**) shows the activity against elastase-1, demonstrating that CDR3 C reduced the substrate hydrolysis by 22%. (****) *p* < 0.0001.

**Figure 9 ijms-25-05181-f009:**
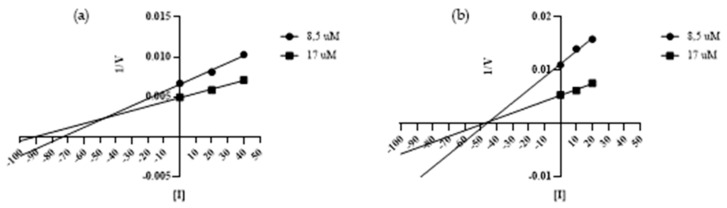
Representative Dixon plot for the characterization of the type of inhibition exhibited by CDR3 A (panel **a**) and CDR3 E (panel **b**) against thrombin (300 ng/µL) using the Abz-FRSSRQ-EDDnp substrate. Lines indicate a competitive mechanism of action. Kis of 61.77 µM (CDR3 A) and 55.75 (CDR3 E) were obtained. For the construction of the graph, the software GraphPad Prism 8 was used.

**Figure 10 ijms-25-05181-f010:**
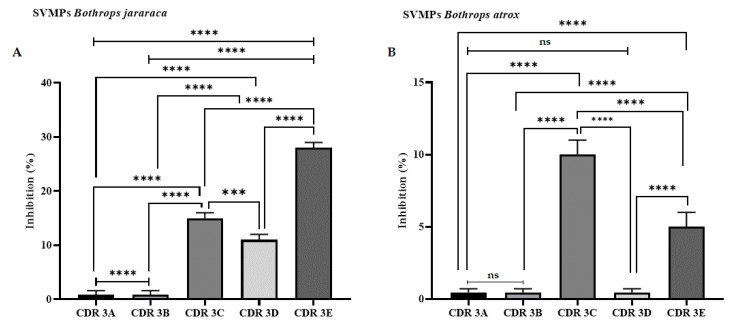
Inhibitory activity assay against metalloproteases (SVMPs) present in *B. jararaca* (**A**) and *B. atrox* (**B**) venoms. The experiments were performed in a fluorimeter (Hidex 425-301, Turku, Finland), with readings adjusted for excitation and emission at 320 and 420 nm using Abz-FASSAQ-EDDnp as a substrate. (***) *p* < 0.001 and (****) *p* < 0.0001. ns: not significant.

**Table 1 ijms-25-05181-t001:** The number of identified peptides in the proteome, from light and heavy chains, digested separately with trypsin and chymotrypsin, based on the three PEAKS approaches described in the methodology.

	Heavy Chain		Light Chain	
PEAKS/Enzyme	Trypsin	Chymotrypsin	Trypsin	Chymotrypsin
De novo	142,122	85,630	120,332	59,991
PEAKS DB	6181	1505	1761	224
SPIDER	7159	1678	5475	1357

**Table 2 ijms-25-05181-t002:** Relative hydrolysis ratios for the β-chain of insulin and the synthetic peptides CDR3 A, CDR3 C, CDR3 D, and CDR3 E.

Peptide	Relative Hydrolysis Ratio (%)
β-chain of insulin	100
CDR3 A	<0.01
CDR3 C	<0.01
CDR3 D	<0.01
CDR3 E	<0.01

Assay performed on a C-18 Shim-Pack column (150 × 4.6 mm) and a gradient of 20–60% B in 20 min [solvent A (H₂O/0.1%TFA) and buffer B (acetonitrile/solvent A in 9:1 ratio)]. All experiments were performed with incubations at 37 °C for 4 h in 100 mM Tris HCl buffer, 20 mM NaCl, pH 7.4, using 0.1 µg of both venoms and 50 µM of all peptides.

**Table 3 ijms-25-05181-t003:** Prediction of hemolytic peptides.

Peptides	ProbScore	Prediction
CDR3 A	0.49	Non-hemolytic
CDR3 B	0.49	Non-hemolytic
CDR3 C	0.48	Non-hemolytic
CDR3 D	0.45	Non-hemolytic
CDR3 E	0.45	Non-hemolytic
ILPWKWPWWPWRR	0.94	Hemolytic

Indolicidin (ILPWKWPWWPWRR) was used as a positive control. The PROB score is the normalized SVM score, ranging between 0 and 1, where 1 indicates a high probability of being hemolytic and 0 indicates a low probability.

## Data Availability

Data are contained within the article and Appendix A.

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
