# Peer review of "Evaluation of the Inhibitory Potential of Synthetic Peptides Homologous to CDR3 Regions of a Monoclonal Antibody against Bothropic Venom Serine Proteases"

_ijms, 2024, doi:10.3390/ijms25105181_

Round 1

Reviewer 1 Report

Comments and Suggestions for Authors

The use of antivenom in medical facilities has led to significant progress in the treatment of snakebites. However, there have been no significant changes or advancements in the treatment of snakebites using antivenoms. While antivenom is vital in reducing the number of fatalities, there is still a need for innovation and improvements in this area. Therefore, the study performed by Saladini and collaborators brings novel findings and promising avenues for rethinking the current antivenom strategy. Peptides have already been commercialised in pharmaceutical industry as drugs with different applications.  The research is relevant with potential implications, but some points need the attention of the authors before its publication. 

Please find below my suggestions and comments.

Major

1. Please include the units of molecular weight in the SDS-PAGE gels. 

2. Figure 1. The chromatographic profile can be added here. 

3. Figure 2. Fractions 6 and 7 contain serine protease, but also other venom components as evidenced by SDS-Page. What are the other potential venom components based on the molecular weight? The chromatography profile can be also included here. 

4. Please improve the quality of the figure 3. The bands are not clear in the gel.

5. What is the purity of synthetic peptides? This information is essential. 

6. It would be useful to see the electrophoretic profile of both crude venoms to see the abundance of serine proteases. I also recommend using SDS-Page to study the interaction between peptides and crude venoms. 

7. Figure 5. Not complete inhibition was observed for synthetic peptides. Did the authors try higher concentrations? Please discuss this. 

8. Figure 5. It would be more informative to observe the activity level including venom enzyme activity here. The inhibition percentage is not as informative as  the enzyme activity. These values allow better comparison between venoms and peptides. 

9. Lines 197-200. Is this ratio comparable with the doses used in clinical settings? 

10. Authors used the values of Km to infer the type of inhibition. The values of Vmax must be also provided and discussed. 

11. The authors explore the ability of peptides to inhibit other enzymes, such as human thrombin. But I am curious about the systemic toxicity of these peptides. Usually, the toxicity of peptides is assessed using a haemolysis assay. Can the authors perform this in vitro study? If not, I also suggest the use of in silico tools to predict the toxicity.  There are different free available tools to predict the toxicity of peptides to red blood cells. 

12. Authors must discuss the implications of the peptides inhibit other enzymes, not only venom serine proteases. 

13. Did the authors evaluate if these peptides can inhibit snake venom metalloproteases?  Can the authors evaluated this?

14. Section 4.6. Please clarify the exact time point used to calculate inhibition and also enzyme parameters. 

15.The scope of the study matches that of IJMS, but not of this special issue. I suggested assessing metalloprotease activity.

Minor

1. The titles of subsections can be removed in the abstract section. For example, background, methods and so on.

2. Keywords: Please avoid the use of the same words found in the title

3. Some sentences must be supported by references. For examples, lines 35-37 and so on. 

4. Line 41. Bothrops atrox is not only the most medically important snake in the Amazon region of Brazil, but also in other countries, such as Ecuador. Please see some references: DOI: 10.1016/j.toxicon.2021.01.007, DOI: 10.1016/j.ijbiomac.2022.03.111. Thus, this statement can be revised. Including the wide distribution of this snake and these references helped to support the importance of studying this venom for the Amazon region, not only for Brazilian communities. 

5. Table 1. Caption should appear directly above every table.

6. Line 298. Please change , by .

7. Please avoid one-sentence paragraphs. Combine paragraphs with similar ideas or expand it. 

Author Response

Reviewer #1

We wish to thank the Reviewer #1 for the rather appropriate comments on our manuscript. The authors agree with all of them, and did correct and modify the text accordingly, aiming at concision.

  1. Please include the units of molecular weight in the SDS-PAGE gels. 

Answer: We agree, and the units of molecular weight in the SDS-PAGE (kDa) gels in Figures 1 and 3 were included

  1. Figure 1. The chromatographic profile can be added here. 

Answer: This purification step was carried out with a syringe, without the use of a chromatographic system. Therefore, we do not have the requested chromatographic profile. To guarantee the reproducibility of the experiment carried out, we added details in item 4.3 (Hybridoma 6AD2-G5 culture and purification), as described below.

The equilibrium and elution of unretained components were monitored by absorbance measurements. using 20 mM sodium phosphate buffer, pH 7.0 (approximately 20 mL). Afterwards, the mAb was eluted with the elution buffer (100 mM citric acid pH 3, approximately 5 mL). In the vial containing the eluted antibodies, 100 µL of 1 M Tris-HCl pH 9 solution was previously added to maintain the stability of the mAb.

  1. Figure 2. Fractions 6 and 7 contain serine protease, but also other venom components as evidenced by SDS-Page. What are the other potential venom components based on the molecular weight? The chromatography profile can be also included here

Answer: We agree that this information is important. Therefore, we added the possible components of the B. atrox venom still present in this purification step, as described below and, for this, a new reference was added.

In addition to the 42 kDa protein band, it is also possible to observe other components present in fractions 6 and 7 with molecular masses around 40 kDa and 22 kDa, which are, probably, metalloprotease of P-II and P-I classes, respectively (Guércio et al., 2006).

Regarding the chromatographic profile obtained in gel filtration, we inform you that it is available in Supporting information Figure 1.

  1. Please improve the quality of the figure 3. The bands are not clear in the gel.

Answer: We agree that panel B of Figure 3 is unclear. We attempted to improve the definition and clarity of the SDS-PAGE that demonstrates the purification of SVTLE from B. atrox.

  1. What is the purity of synthetic peptides? This information is essential. 

Answer: We agree. As described, the peptides were obtained from the company GenOne biotechnologies, and with purity greater than 95%. Along with the peptides, LC-MS/MS analyzes and HPLC profiles were sent, proving the purity of the molecules.

This information was added in item 4.1 Reagents

  1. It would be useful to see the electrophoretic profile of both crude venoms to see the abundance of serine proteases. I also recommend using SDS-Page to study the interaction between peptides and crude venoms. 

Answer: We agree with the relevance of indicating the abundance of SVTLE in B. atrox venom. However, we would like to inform you that we are continuing this study and, at the moment, we are sequencing the CDR1 and CDR2 regions of the mAB used in the present study. Therefore, we kindly request that this information be included in the next manuscript.

  1. Figure 5. Not complete inhibition was observed for synthetic peptides. Did the authors try higher concentrations? Please discuss this.

Answer: To date, higher concentrations of peptides have not been evaluated. Since the inhibition mechanism was determined to be non-competitive, it is expected that higher concentrations of inhibitor will decrease the hydrolysis rate of the substrate used. However, new experiments must be carried out in order to confirm this possibility. A new sentence was added to the discussion, as described below.

Thus, an enzymatic reaction regulated by a non-competitive inhibitor leads to a less efficient catalysis, regardless of the amount of substrate available, and, therefore, higher concentrations of a non-competitive inhibitor should achieve higher inhibition levels.

  1. Figure 5. It would be more informative to observe the activity level including venom enzyme activity here. The inhibition percentage is not as informative as the enzyme activity. These values allow better comparison between venoms and peptides. 

Answer: We agree that activity levels would be better compared. Therefore, we included the levels of specific activities (UF/min/µg) in Figure 5 and indicated in the legend the specific activities obtained with the venoms, as described below.

The numbers below the CDRs indicate the specific activities obtained in the assays (UF/min/µg), and the specific activities obtained with the B. jararac and B. atrox venoms were 150 UF/min/µg and 145 UF/min/µg , respectively.

  1. Lines 197-200. Is this ratio comparable with the doses used in clinical settings? 

Answer: The information in the SAB leaflet is that 1.0 mL is capable of neutralizing 5 mg of bothropic venoms. Therefore, the doses used in vitro for serum neutralization studies are higher than those used in snakebite accidents. Thus, the doses chosen for our tests were determined experimentally in order to obtain inhibitions with the use of SAB. In addition, the tests were carried out with 30 min of pre-incubation of the SAB with the venoms, which does not happen in clinical settings.

  1. Authors used the values of Km to infer the type of inhibition. The values of Vmax must be also provided and discussed. 

We agree. The Vmax value for the hydrolysis of the Abz-Ser substrate by SVTLE present in B. atrox venom were added and discussed in the text (the attained Michaelis-Menten curve is presented in Supporting Information 2).

  1. The authors explore the ability of peptides to inhibit other enzymes, such as human thrombin. But I am curious about the systemic toxicity of these peptides. Usually, the toxicity of peptides is assessed using a haemolysis assay. Can the authors perform this in vitro study? If not, I also suggest the use of in silico tools to predict the toxicity.  There are different free available tools to predict the toxicity of peptides to red blood cells. 

Answer: We agree. This is very important information. Therefore, before obtaining the synthetic peptides, we performed in silico analyzes to verify potentially hemolytic peptides. The results demonstrated that the 5 peptides were predicted to be "non-hemolytic". This information were added to the Manuscript, as described below.

  1. Results

2.9 In silico analysis for the prediction of hemolytic activity

The possible hemolytic activity of the peptides was analyzed through in silico studies. The results obtained on the HemoPI platform: Hemolytic Peptide Identification Server are shown in the Table 3, indicating that the peptides do not present hemolytic activity.

Table 3 – Prediction of hemolytic peptides

Indolicidin (ILPWKWPWWPWRR) was used as a positive control. The PROB score is the normalized SVM score, ranging between 0 and 1, where 1 indicates a high probability of being hemolytic and 0 indicates a low probability.

  1. Materials and Methods

4.11 Prediction of hemolytic peptides

The HemoPI platform: Hemolytic Peptide Identification Server  (https://webs.iiitd.edu.in/raghava/hemopi/) was used to predict the hemolytic activity of synthetic peptides. Indolicidin (ILPWKWPWWPWRR), described as having broad hemolytic activity, was used as positive control. The “SVM (HemoPI-1) based” prediction method was used for the analysis and the PROB score, ranging between 0 and 1 (where 1 indicates a high probability of being hemolytic and 0 indicates a low probability).

Peptides

ProbScore

Predcition

CDR3 A

0.49

Non-hemolytic

CDR3 B

0.49

Non-hemolytic

CDR3 C

0.48

Non-hemolytic

CDR3 D

0.45

Non-hemolytic

CDR3 E

0.45

Non-hemolytic

ILPWKWPWWPWRR

0.94

Hemolytic

  1. Authors must discuss the implications of the peptides inhibit other enzymes, not only venom serine proteases. 

We agree. A new sentence was added about the importance of expanding our selectivity studies to other proteases, and not just the serine proteases studied so far. See below.

In addition to inhibiting the SVTLE present in the venom of B. atrox, the peptides CDR3 A and CDR3 E also proved to be inhibitors of human thrombin, however, the values of the inhibition constants obtained were about 60 times higher. Metalloproteases are considered important and abundant toxins present in bothropic venoms and, therefore, experiments on the possible interaction with the synthetic CDRs were carried out. Only CDR3 E was able to inhibit the metalloproteases present in B. jararaca venom, when used at a concentration 20 times higher than that used in studies with serine proteases. This is an unexpected result, but the simultaneous inhibition of toxins belonging to the metallo and serine protease classes could be of clinical interest. Although the other serine proteases studied were not inhibited by the synthetic peptides, a broader panel of proteases, in addition to serine proteases, should be studied to verify the selective inhibition of SVSPs.

  1. Did the authors evaluate if these peptides can inhibit snake venom metalloproteases?  Can the authors evaluated this?

Answer: Yes, we have these results. For the assays, we used a FRET substrate that had already been characterized as selective for metalloproteases present in bothropic venoms. Thus, the experimental description and results obtained were added to the manuscript and were also discussed.

Results

  1. Materials and Methods

Results

2.8 Analysis of the inhibition selectivity of synthetic peptides

In addition to studies with serine proteases, assays on the inhibition of metalloproteases present in B. atrox and B. jararaca venoms were carried out. For this, the FRET substrate Abz-FASSAQ-EDDnp (Abz-Metal) was used, since it is selective for metalloproteases from bothropic venoms. The results indicated that CDR3 E was capable of inhibiting the hydrolysis of Abz-Metal by 30%, when using 100 µM of the peptide. In studies with B. jararaca venom. Regarding the results obtained with the B. atrox venom, only CDR3 C was able to reduce substrate hydrolysis by 10%, also when used at a concentration of 100 µM (Figure 10).

Inhibitory activity assay against metalloproteases presents in B. jararaca (A) and B. atrox (B) venoms. The experiments were performed in a fluorimeter (Hidex 425-301 Finland), with readings ad-justed for excitation and emission at 320 and 420 nm using Abz-FASSAQ-EDDnp as substrate. Statistical analysis was performed using the one-way ANOVA method, followed by the Tukey post-test, * p value <0.0001. For the construction of the graph, the software GraphPad Prism. 8 was used.

  1. Materials and Methods

4.11 Evaluation of the inhibition of metalloproteases from bothropic venoms by synthetic CDRs

The assays were performed in 100 mM Tris-HCl buffer, 20 mM NaCl, pH 7.4 (final volume 100 µL), using the 96-well plate (Corning) and the Abz-FASSAQ-EDDnp substrate (5 µM). All reactions occurred at 37 ºC using a concentration of 50 ng of B. jararaca venom and 20 ng of B. atrox venom. The assays were performed on a Hidex fluorimeter (425-301 Hidex, Finland), with excitation and emission fluorescence set at 320 and 420 nm, respectively. Readings took place for 15 min, with 1 min intervals. Experiments were performed in triplicate.

  1. Section 4.6. Please clarify the exact time point used to calculate inhibition and also enzyme parameters. 

Answer: We agree. Information was added to section 4.6 aiming for clarity and reproducibility of the experiments.

15.The scope of the study matches that of IJMS, but not of this special issue. I suggested assessing metalloprotease activity.

Answer: We totally agree. We hope that the addition of results obtained with metalloproteases (question 13) can resolve this flaw.

Minor

  1. The titles of subsections can be removed in the abstract section. For example, background, methods and so on.
  2. Keywords: Please avoid the use of the same words found in the title
  3. Some sentences must be supported by references. For examples, lines 35-37 and so on. 
  4. Line 41. Bothrops atrox is not only the most medically important snake in the Amazon region of Brazil, but also in other countries, such as Ecuador. Please see some references: DOI: 10.1016/j.toxicon.2021.01.007, DOI: 10.1016/j.ijbiomac.2022.03.111. Thus, this statement can be revised. Including the wide distribution of this snake and these references helped to support the importance of studying this venom for the Amazon region, not only for Brazilian communities. 

Answer: We agree. Reference #5 used in the first version of the manuscript was replaced by Patiño, R. S. P. , Salazar-Valenzuela, D. , Medina-Villamizar, E., Mendes, B., Proaño-Bolaños, C.,  Silva, S. L.,  Almeida, J. R. 2021. Bothrops atrox from Ecuadorian Amazon: Initial analyses of venoms from individuals. Toxicon. doi: 10.1016/j.toxicon.2021.01.007

  1. Table 1. Caption should appear directly above every table.
  2. Line 298. Please change , by .
  3. Please avoid one-sentence paragraphs. Combine paragraphs with similar ideas or expand it. 

Answer: We agree. All items were met as requested.

Reviewer 2 Report

Comments and Suggestions for Authors

Saladini et al. present a prospective, in vitro investigation of the efficacy of newly synthesized complementarity-determining regions (CDRs) modeled after key portions of antibody molecules in a Bothrops antivenom (BAV) that recognize snake venom thrombin like (SVTLE) venom serine proteases. A variety of small molecular weight peptides that were CDR’s were tested against the enzymatic activity of B. atrox B. jararaca venoms and SVTLE with a variety of milieus. The CDRs’ ability to inhibit human thrombin, trypsin, and elastase 1. The rationale for this investigation is to develop a more focused, small molecular weight antivenoms to deal with the effects of a variety of Bothrops species’ venom that may not be well covered by the administration of BAV.

The authors meticulously document the isolation and synthesis of the key molecules utilized for experimentation. The antibody molecule of interest was antithrombin-like, which is why testing synthetic CDRs-like peptides (CDR’s hereafter in this review) derived from it for inhibition of human thrombin, etc. was critical. The results presenting the purification of the antibody molecule and the modeling of CDRs from the key parts of the molecule was well presented. Purification of the SVTLE and MS description of digestion of the antivenom molecule are also well presented. The methodical presentation of the inhibitory effects of various CDRs on raw venom and use of Michaelis-Menten kinetic studies to define a non-competitive mechanism of inhibition by the CDRs was very interesting. This is consistent with what would be expected from an antithrombin-substrate interaction.

It was interesting and expected that some of the CDRs did inhibit human thrombin and the other two non-snake venom enzymes. What was not expected was that this inhibition was competitive – aka, reversible. This is a bit unexpected and fortunate, as the propensity for bleeding should be less problematic in humans. This should be addressed in Discussion.

I have only a few comments.

1. Illustrations. Please increase the size of the graphics or consider stacking the panels vertically and enlarging them so that the units on the axes are more easily read.

2. Statistics. Please create a statistical section in Methods. That way the methods and vendor of the software only needs to be presented once instead of in each figure legend. Please do this for the kinetic studies as well.

3. Statistical power. It is difficult to believe that three replicates across six different conditions provides a statistical power equal to 0.8. I understand that there is significance, but even with Tukey I am curious about the power. Your statistical output will have the value.

In summary, this is an interesting study that methodically document the creation of efficacious CDR’s with state-of-the-art techniques. I look forward to in vivo experiments with these materials in the future.  

Comments on the Quality of English Language

There are a few typographical errors here and there.

Author Response

Reviewer #2

We wish to thank the Reviewer #2 for the rather appropriate comments on our manuscript. The authors agree with all of them, and did correct and modify the text accordingly, aiming at concision. The article was revised and adjustments in the English were made, as highligth in the text.

It was interesting and expected that some of the CDRs did inhibit human thrombin and the other two non-snake venom enzymes. What was not expected was that this inhibition was competitive – aka, reversible. This is a bit unexpected and fortunate, as the propensity for bleeding should be less problematic in humans. This should be addressed in Discussion.

Answer: We agree. Thank you for this observation. A new phrase as added at Discussion. See below.

Furthermore, thrombin inhibition by CDR3 A and CDR3 E presented a competitive inhibition mechanism. Although unexpected, this result can be considered beneficial, as competitive inhibitors are reversibles and the propensity for bleeding should be less problematic in humans.

I have only a few comments.

  1. Illustrations. Please increase the size of the graphics or consider stacking the panels vertically and enlarging them so that the units on the axes are more easily read.

Answer: We agree that the figures should be larger so that all information are better visualized. However, at this point of submission, we are following IJMS rules and hope, if the manuscript is accepted for publication, that the figures are of an appropriate size.

  1. Statistics. Please create a statistical section in Methods. That way the methods and vendor of the software only needs to be presented once instead of in each figure legend. Please do this for the kinetic studies as well.

Answer: We agree. A new section was created to describe the statistical analyzes and the captions were corrected. See below.

The results obtained were statistically analyzed in the GraphPad Prism 8 program, using the one-way ANOVA test followed by Tukey's post-test. Results with a p value <0.05 (*) were considered significant. The statistical power check was carried out using the G* Power 3.1.9.7 software.

  1. Statistical power. It is difficult to believe that three replicates across six different conditions provides a statistical power equal to 0.8. I understand that there is significance, but even with Tukey I am curious about the power. Your statistical output will have the value.

Answer: You are right. To have a power of 0.8 in our analysis, we had to revisit the results and increase the number of experiments to four. Thanks for the observation.

In summary, this is an interesting study that methodically document the creation of efficacious CDR’s with state-of-the-art techniques. I look forward to in vivo experiments with these materials in the future.  

Thank you so much. We are currently sequencing the mAB CDR1 and CDR2 regions with the intention of designing a mimetic peptide for in vivo assays.

Reviewer 3 Report

Comments and Suggestions for Authors

In the present manuscript, the authors address the issue of safe neutralization of snake bites, one of the most challenging health subjects within the scope of global sustainable development goals. They used a supernatant of hybridoma antibody, neutralizing a serine protease-type toxin present within snake venom, to isolate the relevant monoclonal antibody and then performed a protease digest and mass-spectrometry analysis to define the putative sequences of CDR3 loops of the heavy and light chains of the antibody. They have then synthesized the identified CDR3-pertaining peptides, first the ones containing either the heavy or the light chain sequences, and then combined variants connected over a flexible linker, and probed them in neutralization assays with the venom. Further, the peptides were tested for their specific activity on commonly used proteases thrombin, trypsin and elastase-1, which are only distantly related with the enzymes contained within the snake venom.

As much as such contributions are valuable to the development of intervention agents, which could eventually replace serum therapy and thus provide a superior safety, the manuscript must be corrected for several inconsistencies.

Major points:

-          Most importantly, the sequences of the used peptides must be rendered correctly.

-          The language of the manuscript is at points not at an acceptable level, while certain passages are flawless. I strongly recommend a careful re-read by all the authors before the resubmission.

-          In the reviewer’s version, the supplementary data are missing, and should be provided upon the resubmission.

I will only be able to make a judgement on the manuscript data after the first correction, but in the view of importance of the subject, I strongly support the invitation for a revision and the resubmission. Please find below a list of remarks which I hope you will find helpful.

Line 2: CDR3 regions (please correct throughout the text)

Line 64: „Antibody affinities and specificities are established…” affinity and specificity is not determined only by CDRs – I propose to change into: Antibody affinities and specificities crucially depend on six hypervariable loops at the antigen-binding site, or similar

Line 66: “Among these, the portions called CDR3s are considered critical for antigen recognition.” – Maybe reword to something like: The portions called CDR3 are most diverse in their length and composition and therefore considered most decisive for antigen recognition.

Line 73: “The effective purification – the efficient purification

Line 74: anti-thrombin enzyme-like antibody

Line 75: “…of the antibody (6AD2-G5) from the culture medium is observed, with a yield of 0.2 mg/mL”: this is probably the concentration. Please state what amount was obtained from what quantity of hybridoma supernatant.

Figure A1: relevant marker bands should be labeled, what are the two bands above 95 kDa?

Line 86: membranes were developed

Line 123: despite relatively short size of the sequences

Line 137: The assembly resulted in…

Line 141: These were probably the sequences that were identified as CDR3 regions of VH and VL?

Line 156: cysteine residues were appended to the peptides that were synthetized later, but surely for stability there should be another Cysteine residue present (in the immunoglobulin domains, this cysteine is paired with another cysteine residue)? Please explain.

Line 163-164: Both peptides that were newly synthetized are of the same sequence. The section of peptide sequences corresponding to the L-CDR3 now contains an isoleucine reside instead of Leu as it did when it was discovered (I instead of L) – please explain.

Lines 168-169: Again, peptides 3D and 3E have the same sequence.

Line 255: word order: for each substrate concentration a control test in absence of the peptide was performed

Line 257: during all experiments

Line 298: decimal point and not comma should be used.

Line 350-351: the discovered peptides should correspond to the CDR3 of a monoclonal antibody, however for the heavy chain 2 different peptides were found and chosen for further use– please explain.

Line 365: that it is not necessary

Line 374: “were more, or equally, effectives” – were more or equally effective

Line 392: without the need to use serum-based antivenoms.

Line 414: these are probably 2x10e6 viable cells (also in the line 418). What was the volume of the hybridoma culture and were these 2x10e6 cells per mL, or in total?

Line 469: “Mus musculus and Homo sapiens” – species names should be in italics

Line 502: “Data Availability. “ - is here probably by mistake

Line 507: ÄKTA

Line 515: were applied to an anion exchange column

Line 579: were injected analyses?

Line 583: selective inhibition

Lines 600-601: “The association of two peptides using a linker sequence made up prolines and glycines indicated an increase slight in inhibitory efficacy.” – I propose rewording this sentence: When the peptides presenting a putative CDR3 of the heavy and the light chain were joined, the level of inhibition was slightly higher, or similar

Line 616: commas are missing

Reference list: not all are cited correctly, for example 36, 37, 38, 39, 52, parts of authors’ names, title and other items are missing.

Comments on the Quality of English Language

The language of the manuscript is at points not at an acceptable level, while certain passages are flawless. I strongly recommend a careful re-read by all the authors before the resubmission.

Author Response

Reviewer #3

We wish to thank the Reviewer #3 for the rather appropriate comments on our manuscript. The authors agree with all of them, and did correct and modify the text accordingly, aiming at concision. The article was revised and adjustments in the English were made, as highligth in the text.

Major points:

-          Most importantly, the sequences of the used peptides must be rendered correctly.

Answer: We apologize for the error. Sequences have been corrected throughout the text

-          The language of the manuscript is at points not at an acceptable level, while certain passages are flawless. I strongly recommend a careful re-read by all the authors before the resubmission.

Answer: We appreciate the comment. The text has been carefully revised and we hope that the quality of the language has improved in line with expectations

-          In the reviewer’s version, the supplementary data are missing, and should be provided upon the resubmission.

Answer: We appreciate the observation. The supplementary data was submitted according to the platform's instructions and we do not understand the lack of material for the reviewers. The link created during submission is below. In an attempt to resolve this issue, supplementary material was added at the end of the resubmitted manuscript.

New upload (zenodo.org)

I will only be able to make a judgement on the manuscript data after the first correction, but in the view of importance of the subject, I strongly support the invitation for a revision and the resubmission.

We really appreciate the comment and understand your decision. We work hard on correcting the article and we hope that we have reached the quality necessary for its publication

Please find below a list of remarks which I hope you will find helpful.

Line 2: CDR3 regions (please correct throughout the text)

Answer: We agree. Corrections were made throughout the text.

Line 64: „Antibody affinities and specificities are established…” affinity and specificity is not determined only by CDRs – I propose to change into: Antibody affinities and specificities crucially depend on six hypervariable loops at the antigen-binding site, or similar

Answer: We agree. Corrections were made as suggested.

Line 66: “Among these, the portions called CDR3s are considered critical for antigen recognition.” – Maybe reword to something like: The portions called CDR3 are most diverse in their length and composition and therefore considered most decisive for antigen recognition.

Answer: We agree. Corrections were made as suggested.

Line 73: “The effective purification – the efficient purification

Answer: We agree. Corrections were made as suggested.

Line 74: anti-thrombin enzyme-like antibody

Answer: We agree. Corrections were made as suggested.

Line 75: “…of the antibody (6AD2-G5) from the culture medium is observed, with a yield of 0.2 mg/mL”: this is probably the concentration. Please state what amount was obtained from what quantity of hybridoma supernatant.

Answer: We agree. This information has been added no item 2.1 (Purification of the anti-thrombin-like enzyme antibody (6AD2-G5)).

Figure A1: relevant marker bands should be labeled, what are the two bands above 95 kDa?

Answer: We noted inconsistencies in Figure 1. Molecular markers were included in panels A2 and B2. Our hypothesis for the high molecular mass bands (above 260 kDa) observed in Figure A1 is that the formation of complexes between the mABs molecules has occurred. This hypothesis was added to the Manuscript. Please, see item 2.1 Purification of the anti-thrombin-like enzyme antibody (6AD2-G5).

Line 86: membranes were developed

Answer: We agree. Corrections were made as suggested.

Line 123: despite relatively short size of the sequences

Answer: We agree. Corrections were made as suggested.

Line 137: The assembly resulted in…

Answer: We agree. Corrections were made as suggested.

Line 141: These were probably the sequences that were identified as CDR3 regions of VH and VL?

Answer: We agree. Corrections were made as suggested.

Line 156: cysteine residues were appended to the peptides that were synthetized later, but surely for stability there should be another Cysteine residue present (in the immunoglobulin domains, this cysteine is paired with another cysteine residue)? Please explain.

Answer: Regarding the 3 peptides called "first generation", Cys residues were added to the N-terminal regions because they are present in the sequenced frameworks, in addition to the stability they provide (Figure 4a). In the 2 second generation peptides, the Cys residues were maintained, but without the formation of disulfide bonds. This information has been added to the text

Line 163-164: Both peptides that were newly synthetized are of the same sequence. The section of peptide sequences corresponding to the L-CDR3 now contains an isoleucine reside instead of Leu as it did when it was discovered (I instead of L) – please explain.

Lines 168-169: Again, peptides 3D and 3E have the same sequence.

Answer: We thank you for the important observation an apologize, once again, for the errors made in the description of the peptide sequences studied. Since Leu and Ile residues cannot be differentiated by the mass spectrometry equipment used, we chose to synthesize the peptides with the Leu residue as they are more abundant in proteins when compared to the Ile residue.

Line 255: word order: for each substrate concentration a control test in absence of the peptide was performed

Answer: We agree. Corrections were made as suggested.

Line 257: during all experiments

Answer: We agree. Corrections were made as suggested.

Line 298: decimal point and not comma should be used.

Answer: We agree. Thanks for the observation. Corrections were made as suggested.

Line 350-351: the discovered peptides should correspond to the CDR3 of a monoclonal antibody, however for the heavy chain 2 different peptides were found and chosen for further use– please explain.

Answer: We agree. This is important information, so the sentence below has been added to the Discussion.

It is important to mention that two sequences corresponding to possible CDR3 portions originating from VH were sequenced by mass spectrometry analyses. Thus, in the uncertainty of the correct sequence, the two peptides were synthesized and named as CDR3 A and CDR3 B. Since CDR3 B did not show interactions with both venoms studied, as well as purified SVTLE, CDR3 A must correspond to the sequence present in the mAB.

Line 365: that it is not necessary

Answer: We agree. Corrections were made as suggested.

Line 374: “were more, or equally, effectives” – were more or equally effective

Answer: We agree. Corrections were made as suggested.

Line 392: without the need to use serum-based antivenoms.

Answer: We agree. Corrections were made as suggested.

Line 414: these are probably 2x10e6 viable cells (also in the line 418). What was the volume of the hybridoma culture and were these 2x10e6 cells per mL, or in total?

We agree. The information was added in item 2.1 Purification of the anti-thrombin-like enzyme antibody (6AD2-G5).

Line 469: “Mus musculus and Homo sapiens” – species names should be in italics

Answer: We agree. Thanks for the observation. Corrections were made as suggested.

Line 502: “Data Availability. “ - is here probably by mistake

Answer: We agree. Thanks for the observation. Corrections were made as suggested.

Line 507: ÄKTA

Answer: We agree. Corrections were made as suggested.

Line 515: were applied to an anion exchange column

Answer: We agree. Corrections were made as suggested.

Line 579: were injected analyses?

Answer: We agree. Thanks for the observation. Corrections were made as suggested.

Line 583: selective inhibition

Answer: We agree. Thanks for the observation. Corrections were made as suggested.

Lines 600-601: “The association of two peptides using a linker sequence made up prolines and glycines indicated an increase slight in inhibitory efficacy.” – I propose rewording this sentence: When the peptides presenting a putative CDR3 of the heavy and the light chain were joined, the level of inhibition was slightly higher, or similar

Answer: We agree. Corrections were made as suggested.

Line 616: commas are missing

Answer: We agree. Corrections were made as suggested.

Reference list: not all are cited correctly, for example 36, 37, 38, 39, 52, parts of authors’ names, title and other items are missing.

Answer: We agree. Thanks for the observation. Corrections were made as suggested.

Round 2

Reviewer 1 Report

Comments and Suggestions for Authors

The authors have critically addressed the main points raised. I do not have any further comments. The changes benefit the quality of the manuscript. Taken all together, the manuscript brings novelty to the field with promising avenues for antivenom discovery and can be published in its current form.

Author Response

We thank the Reviewer for appropriate comments on our manuscript. Without a doubt, his help was essential for the manuscript to reach the appropriate level for publication at IJMS

Reviewer 3 Report

Comments and Suggestions for Authors

The authors have corrected the manuscript and addressed all the reviewer’s questions. They have also notably improved the language used in the paper. I only have few minor remarks:

Line 87: kDa

Lines 98-99: Molecular weight marker is in the first lane and displays bands between 10 and 260 kDa.

From line 255: Table 2 has line numbers inserted.

Line 587: Please check that the references 55 and 56 in this line are correct.

Author Response

We thank the Reviewer for appropriate comments on our manuscript. Without a doubt, his help was essential for the manuscript to reach the appropriate level for publication at IJMS.